# Facial blushing and feather fluffing are indicators of emotions in domestic fowl (*Gallus gallus domesticus*)

**Cécile Arnould**⬡*, **Scott A. Love**⬡, **Benoît Piégu**, **Gaëlle Lefort**⬡, **Marie-Claire Blache**, **Céline Parias, Delphine Soulet**⬡, **Frédéric Lévy, Raymond Nowak, Léa Lansade, Aline Bertin**⬡*

CNRS, IFCE, INRAE, Université de Tours, PRC, Nouzilly, France

* cecile.arnould@inrae.fr (CA); aline.bertin@inrae.fr (AB)

## Abstract

The study of facial expressions in mammals provided great advances in the identification of their emotions and then in the comprehension of their sentience. So far, this area of research has excluded birds. With a naturalist approach, we analysed facial blushing and feather displays in domestic fowl. Hens were filmed in situations contrasting in emotional valence and arousal level: situations known to indicate calm states (positive valence / low arousal), have rewarding effects (positive valence / high arousal) or induce fear-related behaviour (negative valence / high arousal). Head feather position as well as skin redness of comb, wattles, ear lobes and cheeks varied across these situations. Skin of all four areas was less red in situations with low arousal compared to situations with higher arousal. Furthermore, skin redness of the cheeks and ear lobes also varied depending on the valence of the situation: redness was higher in situations with negative valence compared to situations with positive valence. Feather position also varied with the situations. Feather fluffing was mostly observed in positively valenced situations, except when hens were eating. We conclude that hens have facial displays that reveal their emotions and that blushing is not exclusive to humans. This opens a promising way to explore the emotional lives of birds, which is a critical step when trying to improve poultry welfare.

## Introduction

Faces generate visual information that can be interpreted by conspecifics. This rich source of information has arguably led to the development of face-specialized neural cortical areas in several animal species including humans [1]. Facial expressions defined as movements of the face [2] have been widely studied in human and non-human mammals. They play a key role particularly in intra-specific communication. According to de Waal [3] "facial displays likely evolved in tandem with the ability to decode them so as to infer the emotional state of others". Facial expressions provide information both on the internal state of the sender (expression of the emotional experience) and its intent to engage in a particular behaviour [3–6]. As a

**Data Availability Statement:** The data and code supporting the findings in this paper are available from Zenodo (https://zenodo.org/doi/10.5281/zenodo.10262049).

**Funding:** This work was supported by INRAE: UMR PRC (CA, SAL, FL, RN, LL, AB) and Métaprogramme SANBA - RED project (CA, SAL, MCB, FL, RN, LL, AB). DS thesis is supported by INRAE and Région Centre Val de Loire. The funders have no role in the data collection and analysis, decision to publish, or preparation of the manuscript.

**Competing interests:** The authors have declared that no competing interests exist.

consequence, emotional expressions have adaptive functions since they participate to the regulation of the interactions between conspecifics.

Studies of facial expressions have led to substantial advances in the understanding of the emotional experiences of nonhuman mammals and how these expressions are involved in communication between conspecifics [7]. Facial expressions have been described and studied in numerous species: primates but also more recently domestic mammals such as dogs, cats, horses [8], pigs [6] and mice [9]. However, facial expressions as a means to indicate emotional experiences may not be a mammal specificity. Despite the reservation of some scientist due to the lack of studies (e.g. [7]), birds could be concerned as well. As mentioned by Emery and Clayton [10] birds have the means to produce facial expressions using head crest or facial feathers. However, the movement of facial feathers has rarely been studied except in a few bird species such as the red-vented bulbul [11]. Yet, identifying bird facial expression would provide a useful tool for exploring and better understanding how they perceive and react to their social and physical environment. This would increase the knowledge acquired on the adaptive value that facial displays have for the social species. In addition, accessing the meaning of their facial displays would be of great interest to promote appropriate interactions (positive ones) between birds and humans. This is critical when looking at bird welfare, particularly in case of domestic species. It is also an ethical question.

In addition to muscle movements of the face, face colour contributes to emotion expression in humans. Blushing of specific facial areas varies with emotions and this information helps to more readily decode emotions of the sender [12], particularly in the case of confusing muscular expressions [13]. Avian species have high spectral sensitivity, elaborate cones and a rich set of pigment [14, 15]. Many birds are very colourful and feather colours are used as signals by conspecifics in a lot of contexts: reproduction, mate selection, parental care and agonistic situations. The colour of pigmented bare parts of the body such as legs, bills, appendices (e.g. caruncles in red-legged partridge) or eye-rings can change depending on environmental conditions (e.g. feeding conditions [16]). The colour of some of these bare parts reflects individual health status, and therefore might affect mate choice by signalling the quality of the sexual partner [17]. Transient changes of skin colour have also been reported, during intraspecific social interaction or during human interaction [18–20], but have never been studied in depth in relation to emotions. The domestic fowl provides an ideal model to study the capacity to express transient skin colour variations of the face (from pale to red) in relation with emotions. This species possesses exposed bare skin areas on their face. In addition, their face is known to play a key role in communication. Head and neck are of greater importance than the rest of the body in individual recognition [21] and appendices of the face, such as the comb, seem to be an indicator of the ability of an individual to dominate a conspecific during dyadic encounters (e.g. [22]). Furthermore, the emotional behaviour of this farmed species is well documented (e.g. [23–25]).

Our objective was to explore the ability of hens to express facial expressions. The evidence of such a capacity would be a big step to go further into the comprehension of their emotional capacities. This would help to assess their sentience. Furthermore, knowing their emotional experience is a critical step when trying to improve their welfare. To do so, we used a naturalistic approach in hens from two rustic chicken breeds. Indeed, we are convinced like Bekoff [26] that "field research on behaviour is of paramount importance for learning more about animal emotions, because emotions have evolved in specifics contexts". We observed feather position and skin colour of the face while hens expressed their daily routine behaviours (resting, feeding, preening, dustbathing, and alert in response to a frightening stimulus) and during two test situations (rewarding and capture test). These seven contexts were contrasted in both emotional valence (positive/pleasant or negative/unpleasant emotions) and arousal level (high or

low intensity), two dimensions that constitute a functional framework widely used to study human and non-human animal emotions [27]. Our hypothesis was that hens express facial expression using the feather position and the colour of their face. We expected that head feather position and the colour of their face varied depending on their current situation, i.e. their current emotional experience. We also wanted to determine if similarities in feather position, or in the colour of their face, would exist in the situations of similar emotional valence and arousal level.

## Material and methods

### Ethics

Behavioural observations are not considered as experimentations and are beyond the scope for ethical consideration regarding French and European animal experimentation regulations. Therefore, the Ethics Committee for Animal Experimentation of Val de Loire (CEEA Vdl, CEEA—019), considered that ethical approval was not required for this study (Protocol number: CE19–2022–1310–1).

Animals investigated were commercial female chickens reared in free range condition. Farmers provided free access to their farm to perform our observations. After the experiment, all hens were sold or given to an association.

### Animals and sites

We investigated two groups of female chickens (*Gallus gallus domesticus*): ten Pekin bantam reared in a commercial farm (Le Haut Montmartre élevage, 37340 Cléré-les-pins, France) and eight Meusienne, a local French breed, reared by a private breeder (M. Audureau) involved in the conservation of local breeds (Chanteraine, 37800 Sainte Maure de Touraine, France). These hens hatched on the farm and were reared in groups of mixed sexes and breeds before the experiments. Only ten female Pekin bantam from the same age and only 8 females Meusienne were available for the experiment. For both breeds, the experimental group was constituted one week before the beginning of the experiment. Pekin (P-hens) and Meusienne (M-hens) are two breeds known for their calmness and closeness with humans [28, 29]. Furthermore, these breeds are more appropriate to study skin colour variations of the face than breeds highly selected on production criteria. Indeed, some breeders selected these later based on the comb shape and colour. Hens were studied in their rearing environment and care was provided by their owner: M. Leduc for P-hens and M. Audureau for M-hens. They were 2 to 3 months-old when filmed for this study. They lived in wooded areas covered with grass (P-hens: 13.3 m x 16 m; M-hens: 25 m x 14.5 m) and had permanent access to a henhouse (P-hens: 1.20 m width, 0.85 m length x 0.70 m height, with access to an additional space 0.35 m x 0.85 m x 0.45 m, henhouse 10 cm above the ground; M-hens: 1.54 m width, 0.80 m length, 0.70 m height in front, 0.40 m height in back, with a platform at the entrance: 0.80 m length, henhouse 35 cm above the ground) containing perches and wood-shavings. Food (mixture of seeds and minerals) and water were provided *ad libitum*. The experiments were performed in summer (late June to august 2020) in sunny weather conditions for both groups.

### Filming of the hens

For two days prior the experiments, hens were familiarized with the observers who remained seated on the ground, and with the camcorders and tripods used later. During this time, observers maintained a calm presence in the flocks and fed hens with mealworms. Hens were confident with the observers during the experiment; they came into contact spontaneously

during the whole duration of the study. To maintain a good relationship with the observers, these later delivered mealworms at the end of the day both in the first days of the experiment and on days where tests were performed.

Each hen was filmed with two camcorders with 4K quality (Sony FDR-AX53; automatic white balance function activated to limit variations that could occur between sunny and more cloudy periods) while conducting several routine behaviours and during two behavioural tests. During daily routines, observers focused on filming four predefined positively valenced behaviours and one negatively valenced: Resting (lying with eyes opened or closed and sometimes pecking the ground), Feeding (eating from the feed trough or grass), Dustbathing (hens squatting in dusty areas of the ground, doing vertical wing shaking, scratching, bill raking, head rubbing, side lying and side rubbing; preliminary phase of pecking and scratching was excluded), Preening (feathers manipulation with the beak, excluding sequences following a dustbathe), and Alert (immobile posture with raised neck following a sudden event: common buzzard in the sky, emitting vocalisations in most cases; dog barking; tractor noise; plane noise passing by at low altitude; alert vocalizations of other fowl in the vicinity). At least twenty minutes of film in total was collected for each routine behaviour and hen. With the exception of Alert, we have excluded all cases where interactions with conspecifics were observed. The observers waited for the specific routine behaviours to be expressed rather than filming a single animal for a given time. This method had the advantage of minimising the movements of the observers and therefore disturbances of the birds. It also allowed filming only when the hen's head was clearly visible. Each routine behaviour (situations) was filmed both in the morning and the afternoon and across the successive days of the observational period. The two behavioural tests were performed to enrich the situations studied. Capture test was negatively valenced and Reward test positively valenced. During the Capture test, each hen was captured as calmly as possible by the same observer. It was the first time they were captured by the observer. When necessary, the hen was gently led into a corner of the field to be captured. The hen was then carried to a specific location of the enclosure to be filmed. They were held in the hands of the observer approximatively 1 meter above the ground for 1 minute (wings were maintained against the hen's body to prevent movements). Each of the right and left profiles of the hen's head was filmed for 30 seconds. After one hen had been filmed, another one was captured. The delay to capture the next hen varied from 1 to 4 minutes in P-hens (mean = 2 min 28 s) and 2 to 5 minutes (mean = 3 min 30 s) in M-hens. During Reward test, hens were filmed eating a highly appealing food, i.e. mealworm. They were individually tested inside their usual enclosure in an arena delimited by wire-mesh (P-hens: l = 89 cm, w = 89 cm, h = 56 cm; M-hens: 118 cm x 59 cm x 58 cm) covered with cardboard (h = 33 cm) to avoid visual contact with conspecifics. A transparent glass dish (Ø = 7 cm, h = 4 cm) containing about 30 mealworms covered with litter was placed inside the arena. All hens had free access to the arena from approximately 36 hours before being tested. They had also been previously familiarized to eat mealworms from the dish inside the arena. On the day of the test, hens spontaneously entered the arena, which was immediately closed. All birds were waiting at the entrance of the arena. It took less than 70 minutes to test all P-hens and less than 50 minutes to test all M-hens. Two observers each filmed the head of the hen for 150 seconds.

Each group of hens (P-hens and M-hens) was filmed by two observers over a four-week period. Films were performed between 10:00 and 18:00 to avoid sunrise and sunset (at this season the sun is high in the sky and temperature variations are limited; the sun rises at 6:00–6:40 and sets at 21:25–22:00, zenith at 14:00). The outdoor temperatures were approximately 25˚C (P-hens) and 27˚C (M-hens) during the video recordings, and the weather was dry and mainly sunny. The first two weeks focused on filming routine behaviours. The behavioural tests were conducted during the third week to obtain hens highly familiarized with the observers. They

were repeated twice to be sure to obtain sufficient data (images with complete head profile): the same day (morning and afternoon) for Capture tests, in the morning of two different days for Rewarding tests. Capture tests were performed first. The fourth week was used to complete films focused on routine behaviour difficult to obtain (mainly feeding and dustbathing sequences). Hen identification was based on physical characteristics (colour, comb shape and size), but also on coloured rings put on one leg (before the experiment) in case of the M-hens because their plumage colourations were very similar.

The behaviours and tests selected for this study allowed us to observe a wide range of emotional situations. Alert is a response to a perceived danger (e.g. predator such as Common buzzard), and capture and manual restraint by human is a frightening event [23] (Jones 1996). Both situations generate fear-related emotions in domestic fowl. Dustbathing and consumption of mealworms (Reward test) have rewarding effects. Both are highly motivated behaviours extensively studied (and used as rewarding stimulus in case of mealworms) in domestic fowl, looking at behaviour, but also neurobiological substrates in case of mealworms. Domestic fowl work for access to mealworms or to dusty substrate to perform dustbathing, and show positive anticipation behaviours [30–32]. Resting, feeding and preening are behaviours that contribute to the maintenance of the health and for preening also the appropriate structure of the feathers (e.g. for insulation). They are performed in situations of absence of threat, i.e. calm situations, and are often considered to be associated with positive affective states (e.g. [24, 33]). Preening is performed when hens are in a state of low arousal and relaxation (e.g. [25, 34]). Resting is closely related to preening and other maintenance behaviours, likely to use time optimally when there is a low risk of danger [35]. Food consumption has hedonic value [36].

### Image extraction, selection and analysis

From each film, images were automatically extracted every two seconds using the "burst capture" function of the free software GOM Player. We then selected images. First, we manually selected those images containing a profile of a hen with all the unfeathered areas of the face visible: appendices (comb, wattles) and denuded skin of the face (cheeks, ear lobes) (Fig 1A). Only images with uniform light and without direct sun were kept. Selection was only based on these three criteria. Second, on the 7933 images obtained (P-hens: 3563 images, M-hens: 4370 images) we selected at random thirty images for each hen and situation using a python script

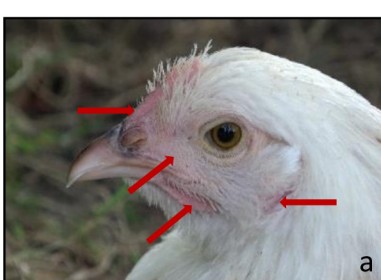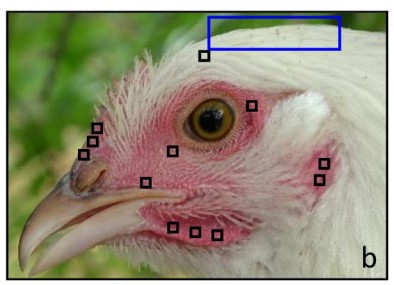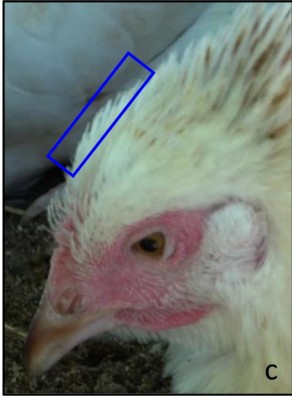

**Fig 1. Profiles of a hen showing different feather positions and skin redness, and the four regions of interest where redness was sampled.** (a) Illustration of the four region of interest (arrows). (b) Locations where skin redness was sampled (black squares, 10 x 10 pixels squares). (b,c) Location where feathers' position was observed (blue rectangle). Figures illustrate Eden (Meusienne hen) with (a, b) feathers labelled as sleeked or (c) fluffed and with (a) low or (b, c) high skin redness. Pictures are extracted from (a) Resting, (b) Capture, and (c) Dustbathing situations.

(https://forgemia.inra.fr/projetred/red_project;sampling_files.py) that avoided, as far as possible, to select images from a same film. As each film corresponded to a hen in a given situation (e.g. a feeding sequence) and films were distributed along the observational period (i.e. several weeks, see above), this method allowed to sample images that came, as much as possible, from different days and behavioural sequences. We sampled thirty images for each hen and situation to minimise any bias that might have occurred if only a few images per situation and per hen had been used since environmental conditions (e.g. lighting and temperature) could not be controlled. For example, it was difficult to calibrate the lighting conditions during filming. Domestic fowl are fast-moving animals with rapid head movements, so it is not possible to place a colour standard close to the head inside the frames. In addition, moving around the animals trying to place colour standard charts would have been disturbing for the hens. We used 1710 images for P-hens (165 to 300 images per situation) and 1336 images for M-hens (112 to 210 images per situation). For some behaviours and hens the number of useable images was less than 30 (S1 Table), however similar results were obtained when analyses were performed on a random sampling of 15 to 30 images per hen and situation.

All images were then analysed by the same experimenter using ImageJ software [37]. Feathers on the top of the head (i.e. crown feathers as defined by Kaplan [38]) were labelled as: fluffed (tip of the feathers visible), sleeked (tip of the feathers not visible) or impossible to decide (Fig 1B and 1C). In addition, as previously performed on blue- and-yellow macaws (*Ara ararauna*; see [20] for more details), we collected the mean value of red (R), green(G) and blue (B) on squares of 10 x 10 pixels located on each region of interest (ROI) and calculated the redness of the skin for each, as R/(R+G+B). This method is used to normalise luminance. The number of squares per ROI varied depending on to the surface visible on the profile analysed. We tried to obtain three squares of 100 pixels each for comb, wattles, cheeks and two squares for ear lobes (area of smaller size; Fig 1B). In order to control for the white balance, we added a square of the same size (100 pixels) on the white feathers in each image (Fig 1B). The median redness values obtained for the white feathers were 0.328 [0.315–0.333] and 0.329 [0.322–0.332] for M-hens and P-hens respectively, quite similar to the 0.333 the value for the pure white standard (R = G = B; S1 Fig).

## Statistical analysis

All data analyses were conducted in R 4.2.2 [39] (R Core Team, 2019).

**Variation of feather position and skin redness of the face depending on the situation.** Feather position. To determine if head feather position varied depending on the situation, we used a binomial Generalized Linear Mixed Model (GLMM) (lme4 package [40]), with Situation as fixed factor and Hen as random factor, followed by post-hoc comparisons with p-value adjusted by Tukey (emmeans package [41]). Images with posturing observations labelled "impossible to decide" and two P-hens with curly plumage were removed from the analysis. As feathers were never fluffed during Reward situation in M-hens, one sleeked and one fluffed value were added to each hen for each situation to avoid convergence problem of the model.

Skin redness. To describe skin redness variations depending on the situation, two analyses were performed. First, we performed a hierarchical cluster analysis (mixOmics package [42]) based on the median redness of the four ROIs. Hen effect was included as random factor. This exploratory analysis was used to identify if some situations were characterized by similar skin redness of the face (situations in a same cluster) and others can be dissociated on this basis, i.e. skin redness of the face (situations located in different clusters). Second, we compared median skin redness between situations for each of the four ROIs. Redness values were log2 transformed to obtain normality of the residuals and homogeneity of the variance assumptions

required for the analyses. The validity of these assumptions was verified using the DHARMa package [43]. We used a Generalized Linear Mixed Model (GLMM) with Situation, ROI and their interaction as fixed-factors and hen as random factor (lme4 package). We then compared situations using emmeans package (p-values adjusted by Tukey) when the factor was significant according to the ANOVA.

**Variation of skin redness of the face depending on arousal and emotional valence of the situation.**   To go further in our analysis on skin redness, we grouped situations with similar arousal or valence based on theoretical frameworks on affective states [27, 33, 41] and literature on poultry behaviour (see above, section Filming of the hens). According to these authors and others, situations that induce fear-related behaviours, have rewarding effect, or calm states are thought to be associated to different affective states characterised by negative valence with high arousal, positive valence with high arousal, and positive valence with low arousal respectively.

Arousal. We compared situations with high versus low arousal. High arousal included Capture, Alert (fear-related behaviours), Dustbathing and Reward (behaviours with appetitive motivational states, i. e. with rewarding effects), whereas Low arousal included Feeding, Resting and Preening (maintenance behaviour, calm states). Even if feeding is considered as rewarding in the literature and as such inducing an emotion with high arousal, we observed very different levels of excitation depending on whether birds were consuming usual non-restricted food or grasses, or highly appetitive food such as insects, mealworms, etc. Furthermore, if both have positive hedonic values, novelty, attention or anticipation are goal-oriented behaviour likely not required in case of *ad libitum* food according to Saper et al [36]. Therefore, we considered that during Feeding arousal level was low compared to Reward, in which mealworms were given.

Valence. We compared situations with 1. negative valence /high arousal (V-/A+; Capture, Alert), 2. positive valence /high arousal (V+/A+; Dustbathing, Reward), and 3. positive valence /low arousal (V+/A-; Feeding, Resting, Preening). Arousal level was considered in association with valence as we observed an effect of arousal level on skin redness. This allowed comparison between situations with similar valence but different arousal levels and allowed access to three affective states: fear-related, with rewarding effect and calm states. For both Arousal and Valence, we used the same model and log2 transformation as previously with Arousal or Valence, ROI and their interaction as fixed factors and hen as random factor.

All data are presented as medians with interquartile ranges. Test significance was considered at $p \le 0.05$

## Results

### Feather position and skin redness of the face varied with the situations

**Feather position.**   In M-hens and P-hens, the percentage of profiles where head feathers were fluffed varied across situation (M-hens: $Chi^2 = 202.0$, P-hens: $Chi^2 = 178.9$; d.f. = 6 and p<0.001 in both cases). In M-hens the percentage of images with feather fluffed was significantly different for Dustbathing, Preening and Resting situations (median $\ge 40\%$ in all three cases) compared to Capture, Alert, Feeding and Reward (0% in all four cases) situations (p < 0.001 in all cases; Table 1). In P-hens results were similar except for Alert. The percentage of images with feather fluffed was significantly different for Alert, Dustbathing, Preening and Resting situations (median $\ge 56\%$ in all four cases) compared to Capture (9%), Feeding and Reward (0% in both cases) situations (p < 0.001 in all cases).

**Skin redness.**   Hierarchical cluster analysis that took into account the redness of the skin of the comb, wattles, ear lobes and cheeks revealed two clusters in both groups (Fig 2A and

**Table 1. Percentage of images where feathers were fluffed depending on the situation.** (a) M-hens. (b) P-hens. A value of 0 means that feathers were sleeked on all the images. Percentages with different letters in a row are significantly different.

| Group | Alert | Capture | Reward | Feeding | Dustbathing | Preening | Resting |
|---|---|---|---|---|---|---|---|
| M-hens (%) | 0 | 0 | 0 | 0 | 40 | 52 | 44 |
| | [0–6] a | [0–59] a | [0–0] a | [0–0] a | [37–66] b | [46–64] b | [33–65] b |
| P-hens (%) | 100 [52–100] b | 9 [0–88] a | 0 [0–2] a | 0 [0–2] a | 56 [21–82] b | 82 [50–100] b | 68 [28–90] b |

2B). Cluster 1 was comprised of situations where hens displayed high redness levels of the ROIs, whereas cluster 2 was comprised of situations where hens displayed low redness levels. Overall, the cluster analyses for M- and P-hens were similar. All M-hens and P-hens in the Capture situation and all M-hens and most of P-hens in the Alert situation were in cluster 1. In contrast, M- and P-hens in the Feeding, Resting and Preening situations were in cluster 2. Hens in the Dustbathing situation were split across clusters 1 and 2. The main difference between groups concerned the Reward situation for which M-hens were in Cluster 1 while P-hens were in Cluster 2.

The ANOVA between full and reduced models revealed significant effects of the two main factors (Situation and ROI) and an interaction effect between the two in both groups (M-hens Situation: $F_{(6, 162)} = 138.73$, ROI: $F_{(3, 162)} = 137.41$, $p < 0.001$ in both cases, Interaction: $F_{(3, 162)} = 2.45$, $p = 0.002$—P-hens Situation: $F_{(6, 243)} = 138.73$, ROI: $F_{(3, 243)} = 122.46$, Interaction: $F_{(18, 243)} = 2.72$, $p < 0.001$ in all three cases). In M-hens (Fig 3A) the redness of the four ROIs (comb, wattles, ear lobes, cheeks) was significantly higher in Alert, Capture and Reward situations than in Feeding, Preening and Resting situations ($p < 0.001$ in all cases). Dustbathing situation had an intermediate position with slight variations according to the ROI. For ear lobes and cheeks, redness in this situation were significantly lower than in Alert,

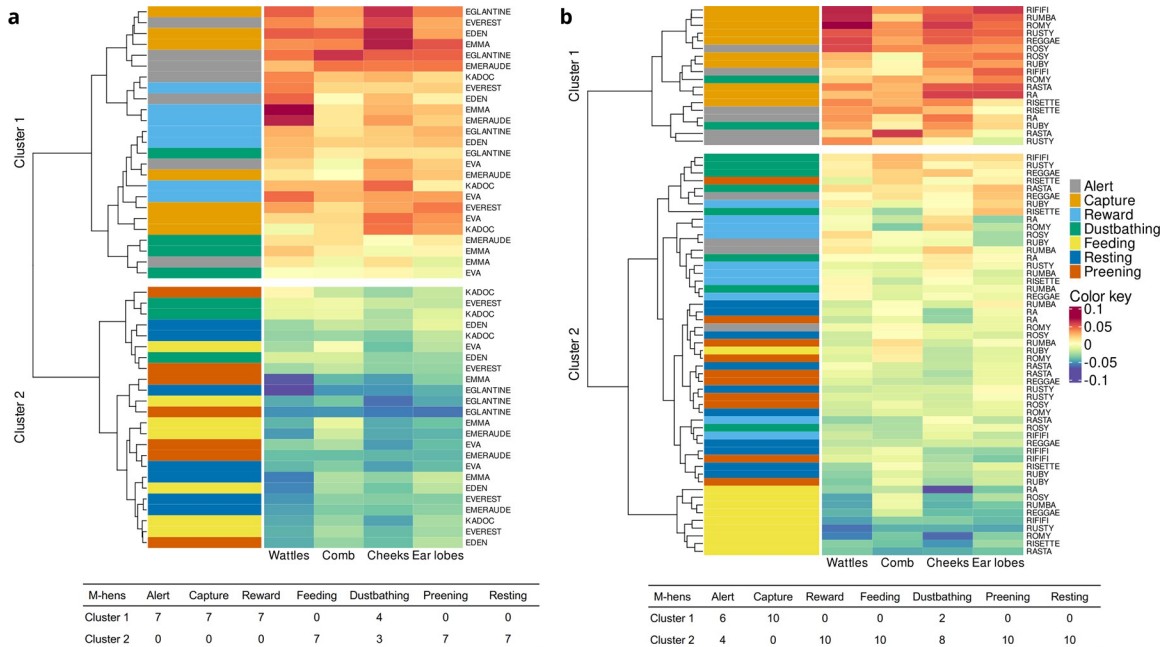

**Fig 2. Dendrogram (left hand side of the figures) and table from hierarchical cluster analyses of the redness of the skin for the four ROIs.** (a) M-hens. (b) P-hens. The heatmap contains the tested situations (Alert, Capture, Reward... in red, blue, green...) in rows and ROI (wattles, comb, cheeks, ear lobes) in columns with red colour indicating high-level of redness, blue colour low-level, and yellow colour being between the two. Tables indicate for each situation the number of hens in each cluster (M-hens, n = 7, P-hens, n = 10).

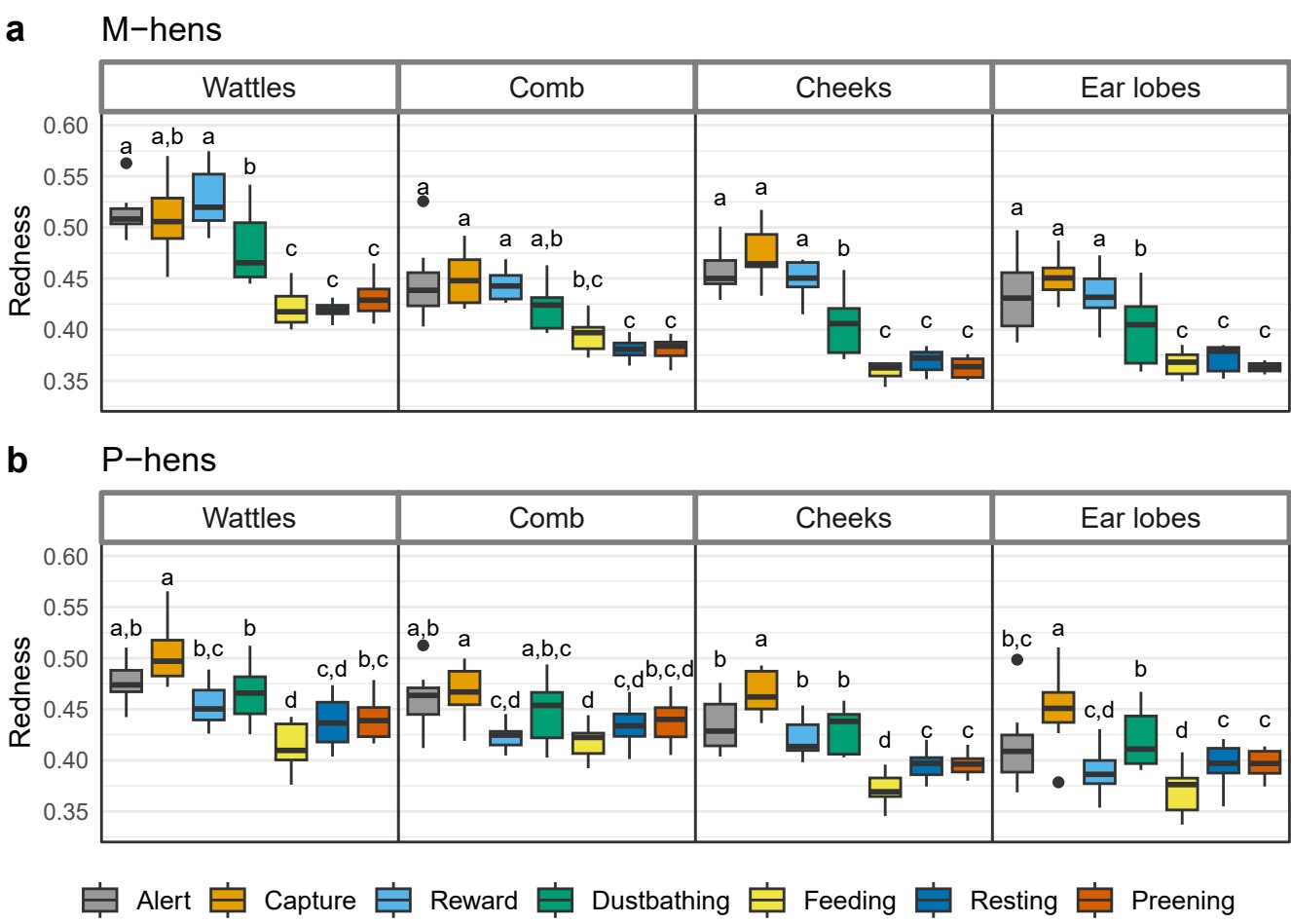

**Fig 3. Median and interquartile range of the redness of the skin for wattles, comb, cheeks and ear lobes depending on the situation.** (a) M-hens. (b) P-hens. Situations with different subscripts differed significantly (GLMM on log2 transformed data with Tukey adjustment for two-by-two comparisons).

Capture and Reward situations and higher than in Feeding, Preening and Resting situations (p < 0.03 in all cases). In P-hens (Fig 3B) cheeks was the ROI that best discriminates the situation and results were quite similar to those observed in M-hens: the redness was significantly higher in Alert, Capture, Reward and Dustbathing situations than in Feeding, Preening and Resting situations (p < 0.016). For the three other ROI, results were similar but situations were more difficult to discriminate statistically, especially for wattles and comb.

## Skin redness varied depending on the arousal and emotional valence of the situation

**Arousal.** The ANOVA between full and reduced models revealed in both groups significant effects of the two main factors (Arousal and ROI) and an interaction effect between the two (M-hens: Arousal $F_{(1, 182)} = 470.56$, ROI $F_{(3, 182)} = 85.07$, p<0.001 in both cases, Interaction $F_{(3, 182)} = 3.93$, p = 0.010 –P-hens: Arousal $F_{(1, 223)} = 149.30$, ROI $F_{(3, 223)} = 60.26$, Interaction $F_{(3, 223)} = 6.11$, p<0.001 in the three cases; Fig 4A and 4B). Hens displayed a higher level of redness during the high- compared to the low-arousal situations in all ROIs (p<0.001 in all cases).

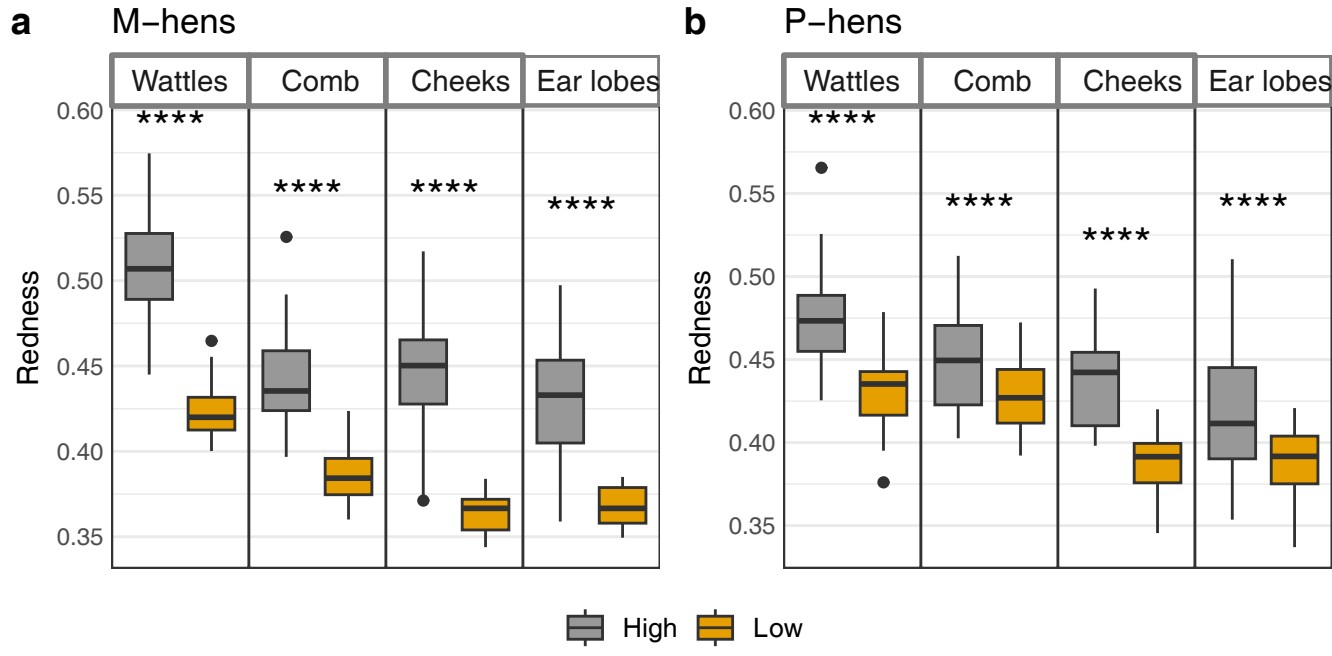

**Fig 4. Redness (median and interquartile range) of the skin for wattles, comb, cheeks and ear lobes depending on the arousal of the situation.** (a) M-hens. (b) P-hens. Low arousal includes Feeding, Resting and Preening; High arousal includes Capture, Alert, Dustbathing and Reward. ****: p<0.001 (GLMM on log2 transformed data with Tukey adjustment for two-by-two comparisons).

**Valence.** In both groups, significant effects of the two main factors (Valence and ROI) and an interaction effect between the two were revealed by the ANOVA (M-hens: Valence $F_{(2, 178)}$ = 293.77, ROI $F_{(3, 178)}$ = 102.53, p<0.001 in both cases, Interaction $F_{(6, 178)}$ = 3.83, p = 0.001 –P-hens: Valence $F_{(2, 259)}$ = 133.43, ROI $F_{(3, 259)}$ = 81.53, p<0.001 in both cases, Interaction $F_{(6, 259)}$ = 3.83, p = 0.001; Fig 5A and 5B). Cheek and ear lobe redness in both groups, and also wattles redness in P-hens, were significantly higher in emotional situations with negative valence (V-/A+) than in emotional situations with positive valence (whether arousal is positive or negative: V+/A+ and V+/A-). Skin redness significantly decreases from V-/A+ to V+/A+ (p ≤ 0.004) and from V+/A+ to V+/A- (p < 0.006 in all cases). For the skin of the comb and wattles in M-hens, no significant difference was observed between negative and positive valence when arousal was high (V-/A+ vs V+/A+; p = 0.148 and p = 0.683 respectively).

## Discussion

Our results show clearly that hens express facial expressions by means of subtle transitory changes in feather position and in the level of skin redness of the denuded areas of their head. Feather position and face redness vary depending on the situation experienced and thus provide useful information on the emotional state being experienced. Specifically, we provide evidence for the first time that female chickens blush in situations of high arousal (negatively valenced, but also positively valenced). Similarity was observed in the facial expression of the two breeds studied, which underlines the robustness of our observations and suggest a functional role of these facial displays.

Crown feather position changed depending on the situation experienced. In situations where hens were eating (Feeding, Reward) feathers were sleeked for almost all profiles as

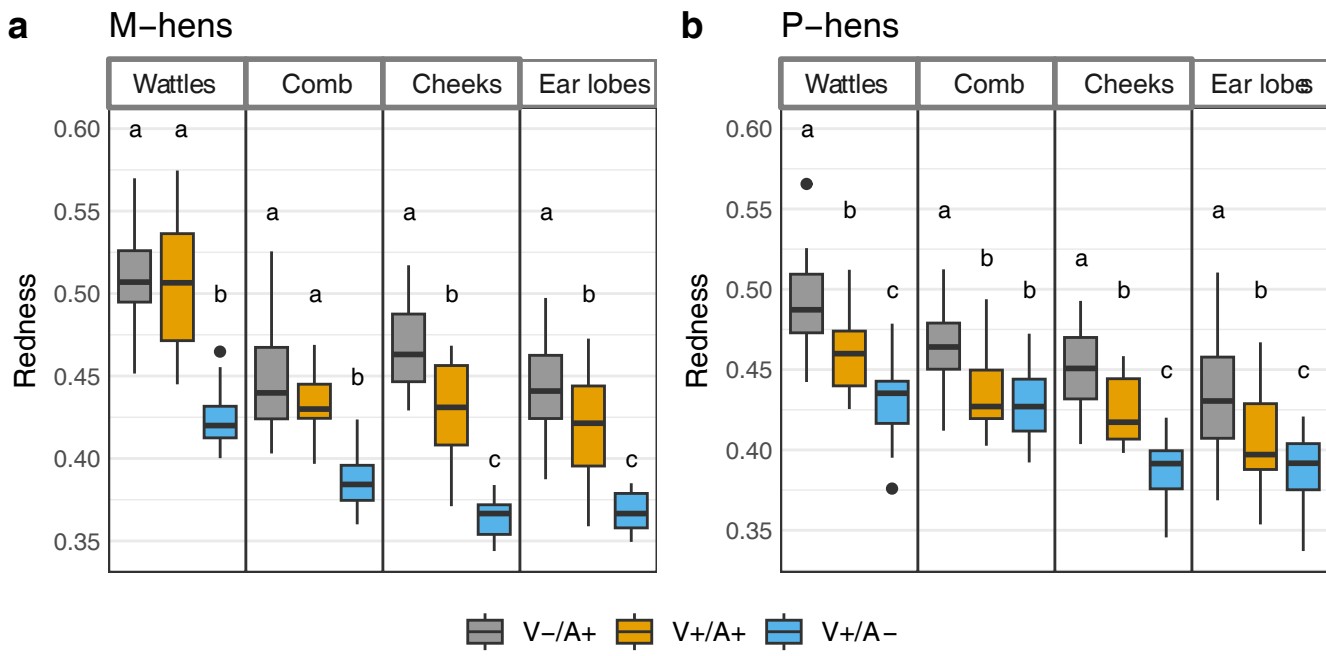

**Fig 5. Redness (median and interquartile range) of the skin for wattles, comb, cheek and ear lobes depending on the valence and arousal of the situation.**
(a) M-hens. (b) P-hens. Negative valence / high arousal (V-/A+) includes Capture and Alert, positive valence / high arousal (V+/A+) includes Dustbathing and Reward, positive valence / low arousal (V+/A) include Feeding, Resting and Preening). Situations with different subscripts differed significantly (GLMM on log2 transformed data with Tukey adjustment for two-by-two comparisons).

observed in blue-and-yellow macaws and sulphur-crested cockatoos (*Cacatua galerita*; [44, 45]). Across the negatively valenced situations, we found varied feather positions. Feathers were observed mainly sleeked during Capture. Interestingly, the only difference in regards to feather position between the groups was observed for the Alert situation in which P-hens displayed a high proportion of profiles with fluffed feathers while M-hens displayed a very low proportion. A difference in the emotional experience between breeds cannot be excluded, but it is more likely that these contrasted facial displays are linked to the biological significance of the situations encountered. The causes of the spontaneous alert behaviour were unpredictable and differed between the two groups. For P-hens alert behaviour was mainly caused by the vocalisations of raptors (also by dog barking or plane noise passing by at low altitude), whereas for M-hens it was initiated by alert vocalizations from conspecifics living in proximity. Several Galliformes species including domestic fowl use different alarm calls in presence of terrestrial or aerial predators [46]. Roosters also modulate alarm call duration as a function of their vulnerability (close to a refuge *versus* out in the open) or of the proximity of conspecifics [47]. As these studies indicate that domestic fowl show variation in perceived risks, here hens might have expressed different facial displays depending on the type of alarming situation. P-hen fluffed position of the feathers might have a camouflage function. When they were in an open area they ran under a tree and looked at the sky, which strengthens this interpretation. If fluffed feathers had a camouflage function, this facial display might be biologically irrelevant in the case of M-hens. These hens were in close vicinity of the vocalizing conspecifics which had caused their alert behaviour. Furthermore, some of them also emitted vocalisations (which was not observed with P-hens). In positively valenced situations such as Resting, Preening and Dustbathing, feathers were fluffed in a high proportion of the profiles (median at 40% or more in M-hens and 56% or more in P-hens). This percentage likely reflects the succession of

different phases (including sometimes pauses of a few seconds) during preening and dustbathing behaviour (e.g. [34, 48]). But also in resting behaviour, since different levels of vigilance occurs (from sparsely pecking on the ground to sleeping with closed eyes). Fluffing of head feathers appears to be a relevant indicator of positive emotional states in situation where birds are healthy. Indeed, feather fluffing could also signal sickness in birds (in this case, birds are often prostrated and feathers are in bad conditions). This facial display can be observed in different situations and species. Indeed, it has been observed during maintenance behaviour, resting and positive social interactions in Psittacidae (macaws and cockatoo) as well as with macaws in presence of their caretaker, or with Japanese quail during dustbathing behaviour [20, 44, 45, 49]. In resting birds this facial display seems to be highly conserved between bird species. Beyond our observation on female domestic fowl, a species highly selected by humans, and Psittacidae, it has been reported in crows (*Corvus frugilegus*; V. Dufour, personal communication), finches, doves and pigeons [50] (Morris 1956). Beyond the temperature function, fluffing of body feathers at rest gives a rounded appearance that stimulates clumping in finches and concentrates alloopreening on the head when only head and neck plumage are ruffled [50]. These behaviours (clumping and alloopreening) facilitate affiliation and group cohesion. Similar social function of feather fluffing probably exists in female domestic fowl. All these analogies (similarities in response to similar selection pressure, as defined by Waller and Micheletta [2] are in favour of an evolutionary continuity between bird species and, as such, could help to elucidate their communicative function.

Our most exciting and informative result concerns blushing and its relation with emotions. The skin redness of the face was lower in situations with positive valence and low arousal where birds are calm and contented (Resting, Preening and Feeding), than in situations with negative valence and high arousal associated to fear-related behaviour (Alert and Capture). Skin redness during situations with positive valence and high arousal that have rewarding effects (Dustbathing and Reward) was intermediate. This suggests a continuum of skin redness providing subtle information about emotional affects rather than the simpler positive/negative valence dichotomy. In case of dustbathing, it seems unlikely that this behaviour had a neutral valence. Birds are highly motivated to dustbathe and according to some authors dustbathing behaviour probably elicits a feeling of pleasure [32, 51]. That said, it is also possible that these intermediate redness values are the result of the dynamics of arousal level during these situations, especially in the case of dustbathing. This behaviour can last more than twenty minutes if birds are undisturbed [52] and has a complex structure, as mentioned above. We observed alternation of phases of high and low redness (as well different level of feather erection), likely due to an alternation of high and low arousal. So far, the mechanism regulating facial blushing in birds is unknown. Anyway, it is unlikely that the higher redness observed in the fear-related behaviours resulted from differences in physical exertion compared to rewarding situations. Indeed, the hens were more active during the dustbathes or reward tests than during the capture or alert situations where they were motionless. Furthermore, redness was low during preening and eating, where hens were physically active, in the same way as during resting.

Cheeks and ear lobes appear more relevant to reveal emotions in female chickens than comb or wattles. These denuded regions, provide the clearest information for distinguishing between valence and arousal levels in both breeds. While the redness of all the ROIs changed with the arousal level (high arousal being associated to higher redness), only the redness of cheeks and ear lobes varied depending on the valence of the situation. In situations with high arousal, redness was highest in situations with negative valence than in positive ones. The different regions of the face might have different communicative functions for the hens. Comb and wattles increase with sexual maturation both in male and in female chickens. Comb and ornament size and colour communicate information on, or are associated with mating success,

dominance, sperm quality or health status in several avian species (e.g. [22, 53–57]). The physiological characteristics of these denuded skin areas might be different from that of cheeks and ear lobes. Comb and wattles might be less appropriate than cheeks and ear lobes to communicate emotional states to conspecifics, even in juvenile hens for which comb and wattles are not fully developed and coloured. This hypothesis requires further evidence from adult hens. Our results should be considered as a first step because they were obtained on two groups of relatively small size. However, we have already replicated the data on the two breeds studied in this experiment (Pekin, Meusienne) in another breed (Sussex, [58]), which reinforce the possibility generalising our results.

Some limitations about our results on blushing can be raised due to our methodological approach. The naturalistic approach we used makes the precise calibration of light conditions during filming difficult. At this date we did not find good solution to insert standard charts inside film without interfering with bird behaviour. Indeed, our method is based on studying free living birds and our objective was to restrict as far as we can the perturbation induce by filming. It is why we choose to measure redness on the white head feathers on each image to validate our method. Variations observed on white feathers showed that colours on images were quite well balanced since we find a redness of 0.33 (which indicate that R, G, and B values for white feathers were similar). Furthermore, the variability observed for white feather redness was far below the variations observed between the situations tested when significant differences were obtained. The naturalistic approach also makes difficult to control with precision for temperature, humidity or wind; it is why we worked on a lot of images (30 images spread along the observational period). This allows to consider the variability that could have been induced by the environmental conditions. However, temperature did not vary very much during our study and our results cannot be explained by thermoregulation variations. Indeed, the blushing measured in our study appears in few seconds and is transitory. It lasts a few seconds or minutes depending on the duration of the stimulus inducing the reaction (a video can be seen on S1 Video), contrary to temperature variations during a day. According to Ioannou et al (2014; [59]), "thermal signal development as a result of vascular change, or muscular activity", measured using thermal infrared imaging, "is rather sluggish compared with other measures of physiological arousal". Facial surface temperature measured by infrared thermography varied with contrasted thermal environments (10, 20 and 30°C; [60]). However, such variations of the temperature did not exist during our observations. Furthermore, it seems that the skin redness we measured and skin temperature measured by infrared thermography do not measure the same phenomenon. Indeed, while we obtained high redness in Reward and Capture situations, the comb temperature dropped in laying hens eating mealworms [61] or in response to handling [62]. Neither can blushing be explained by variations of body temperature associated with activity, as mentioned above. Whatever these limitations, this study brings new knowledges to the understanding of expression of emotion in free living hens. It is the first to assess blushing in birds in such a variety of affective states. The significance of this blushing to regulate interactions between hen rest to be investigated.

Some potential sources of uncertainty exist in the interpretation of our results. First, the two-dimensional model used in the present study that classified emotions depending on valence and arousal is useful to decode emotions, but like every model it has some limitations. Emotions are not fixed, they are part of a dynamic process. For example, motivation to obtain a reward can be followed by the satisfaction of having obtained it. Then, if the emotion is pleasant in both cases the arousal level evolved from high during pursuing and achieving the reward to low when the reward has been obtained. When the goal has been achieved, a phase of contentment appears. Such a dynamic seems to exist during dustbathing for example. Arousal level might vary during this long-lasting behaviour constituted of different phases. The ability

to analyze a high number of images on the sequences filmed would help to capture with more precision the dynamics of the emotion experienced, but adequate methods are currently lacking. Second, the combination of feather position and skin redness of the face likely help to interpret expressions, as the colour of the human face when added to facial-muscular features. Indeed, the colour of the human face facilitates the disambiguation of emotion in the case of confusing facial-muscular expression, such as in early dynamic sequence of anger or disgust [13]. Due to the impossibility to measure feather erection with a nominal scale on the images obtained in the present study, it was impossible to combine these data with the skin redness of the face in a statistical model. However, Benitez-Quiroz et al [12] showed that human emotions can be decoded using facial colour alone, without the involvement of facial musculature. Third, communication is multi-modal [63]. The posture or vocalizations emitted probably facilitate the interpretation of the facial expressions by conspecifics. As the birds observed lived in a group and in a very rich auditory environment, we were unable to discriminate properly the vocal signals emitted simultaneously with facial expressions. Identifying with certainty the vocalizing individuals was not possible.

The characterisation of facial expression during activities associated to low arousal positive state of contentment (low redness and crown feather fluffed) is of great interest for studies on bird welfare. This facial display indicates that birds are calm and secure, a state that enables affiliation and attachment, but also well-being in humans [64]. Decoding this type of emotion is of particular interest in domestic fowl, a species which lives in close contact with humans, and more broadly for all bird species interacting with humans. Descovich et al [7] claims that facial expression measurement could be very useful to assess welfare in mammals. Our results show that birds, and particularly domestic ones, should clearly not be excluded from research on facial expression of emotions, especially by researchers working on bird welfare and searching for indicators of positive emotions. Indeed, such indicators are currently sorely lacking. These commercially important species are in dire need of a better understanding of how they experience and express both positive and negative emotions. This study constitutes a first step in the development of a methodology that could be used in the future to obtain reliable indicators of emotions.

## Conclusions

We describe for the first-time facial expression in domestic fowl by means of variation in crown feather fluffing and skin redness. Domestic fowl showed fluffing, blushing or both in specific emotional contexts varying in both valence and arousal levels. We show that blushing of the face is observed in a domestic avian species, which emotional and cognitive capacities are likely underestimated. The ability to use facial skin redness to infer the emotional state opens a wide window to being able to better understand the emotional experiences of birds. Furthermore, the evidence of blushing in birds experiencing positive and negative emotions, as it is observed in humans, is in favour of an evolutionary continuity in this expressive system between species with sophisticated social interactions. The assumption of Darwin that blushing is "the most human of all expressions" (1872) still highlighted nowadays (e.g. [65]) is seriously weakened by our results, which question the phylogenetic origin of emotional facial blushing in the animal kingdom.

## Supporting information

**S1 Table. Number of images per situation and hen for M-hens and P-hens.**
(DOCX)

**S1 Fig. Redness of white feathers per situation.**
(DOCX)

**S1 Video. Illustration of a rapid blushing following a sudden noise in a Pékin hen.**
(MP4)

## Acknowledgments

We thank Michel Audureau and Manuela Leduc who kindly provided access to their birds in their living environment (M-hens and P-hens respectively) and take care of them, and Jean-Claude Périquet who provided the eggs of the Meusienne hens.

## Author Contributions

**Conceptualization:** Cécile Arnould, Scott A. Love, Frédéric Lévy, Raymond Nowak, Léa Lansade, Aline Bertin.

**Data curation:** Cécile Arnould, Benoît Piégu, Gaëlle Lefort, Marie-Claire Blache, Frédéric Lévy, Aline Bertin.

**Formal analysis:** Cécile Arnould, Benoît Piégu, Gaëlle Lefort, Marie-Claire Blache, Aline Bertin.

**Funding acquisition:** Cécile Arnould, Aline Bertin.

**Investigation:** Cécile Arnould, Benoît Piégu, Gaëlle Lefort, Marie-Claire Blache, Céline Parias, Frédéric Lévy, Aline Bertin.

**Methodology:** Cécile Arnould, Benoît Piégu, Gaëlle Lefort, Marie-Claire Blache, Aline Bertin.

**Project administration:** Cécile Arnould, Aline Bertin.

**Resources:** Cécile Arnould, Benoît Piégu, Gaëlle Lefort, Marie-Claire Blache, Aline Bertin.

**Software:** Cécile Arnould, Benoît Piégu, Gaëlle Lefort, Marie-Claire Blache, Aline Bertin.

**Supervision:** Cécile Arnould, Aline Bertin.

**Validation:** Cécile Arnould, Benoît Piégu, Gaëlle Lefort, Aline Bertin.

**Visualization:** Cécile Arnould, Gaëlle Lefort, Delphine Soulet, Aline Bertin.

**Writing – original draft:** Cécile Arnould, Aline Bertin.

**Writing – review & editing:** Cécile Arnould, Scott A. Love, Benoît Piégu, Gaëlle Lefort, Marie-Claire Blache, Céline Parias, Delphine Soulet, Frédéric Lévy, Raymond Nowak, Léa Lansade, Aline Bertin.

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
