## [Decision Letter · Decision Letter 0]

15 Apr 2024

PONE-D-23-42910Facial blushing and feather fluffing are indicators of emotions in domestic fowl (Gallus gallus domesticus )

PLOS ONE

Dear Dr. Arnould,

Thank you for submitting your manuscript to PLOS ONE. After careful consideration, we feel that it has merit but does not fully meet PLOS ONE’s publication criteria as it currently stands. Therefore, we invite you to submit a revised version of the manuscript that addresses the points raised during the review process.

We look forward to receiving your revised manuscript.

Kind regards,

Birendra Mishra, DVM, PhD

Academic Editor

PLOS ONE

Journal Requirements:

Reviewers' comments:

Reviewer's Responses to Questions

**Comments to the Author**

1. Is the manuscript technically sound, and do the data support the conclusions?

Reviewer #1: Yes

Reviewer #2: Yes

Reviewer #3: Partly

2. Has the statistical analysis been performed appropriately and rigorously? 

Reviewer #1: Yes

Reviewer #2: Yes

Reviewer #3: Yes

3. Have the authors made all data underlying the findings in their manuscript fully available?

Reviewer #1: Yes

Reviewer #2: Yes

Reviewer #3: Yes

4. Is the manuscript presented in an intelligible fashion and written in standard English?

Reviewer #1: Yes

Reviewer #2: Yes

Reviewer #3: Yes

5. Review Comments to the Author

Reviewer #1: The authors addressed the assessment of emotions in different contexts (valence positive or negative /arousal low or high) in domestic fowl by using new behavioural indicators in domestic birds: facial blushing and feather fluffing.

The study is creative and shows interesting results which might be a first step to find applications for improving to assess welfare.

The paper is well written, so apart from minor comments detailed below.

Comments in details :

No key words?

ABSTRACT:

Few information about results for feather fluffing. Is it possible to add little bit more?

INTRODUCTION

L47: Replace our with the.

L61: Could you modify the sentence? The question is more that ethical question, is a welfare question.

MATERIAL and METHODS

You studied 2 strains : Pekin and Meusienne with respectively 10 and 8 hens. Is it a sample of hens within each flock? if it is the case need to be mentioned and add information how you selected the hens.

L121 “ ..known for their calmness and closeness with humans.” add bibliographic reference.

I am not sure to understand how you proceed to organize your sequence for filming.

Did you decide to follow one specific animal? Did you wait for the specific routine behaviours?

Need more information about your method.

L162: “hens spontaneously entered the arena”. No training or motivation stimulus to lead the animal to go in the arena? you wait all the day each animal entered in the arena? What happen with a hen which doesn’t to go inside.

Add information about that.

L150: hens run fast, they didn’t try to escape when you tried to catch them ? How long did you wait before to catch the next one? each capture can have an impact on the all flock.

L174 No rings for P-hens?

Image extraction, selection and analysis:

For some animals and situations, you got few images. What could be the impact on your results?

L198: “Images came, as far as possible, from different days and behavioural sequences”. What does it mean more information.

At which time the image is selected for each behaviour ? at the beginning, in the middle , at the end ?

For example dustbathing last quite a long time and I guess the redness progessed over the time.

L271 : put ad libitum in italic

RESULTS

L291-292: The chi² analyze is not mentioned in statistical section. Please mention it.

L324 -337: Please write a sentence for the 3 others ROI in different situations.

REFERENCES

L715: please correct this reference.

Reviewer #2: This is a very interesting article investigating the redness of skin as a measure of valence and arousal in chickens. My edits and comments are detailed by line number below.

Abstract – Please add at least one statement about the results related to feather position and whether that was or was not repeatably indicative of affective state and why.

LINE 22: Please revise “domestic hens” to “female chickens” here and throughout. “Hen” can refer to females from many different domesticated poultry species.

LINE 84: See LINE 22.

LINE 88: Please revise “rustic hens” to “hens from two rustic chicken breeds”.

LINE 128: What were the outdoor temperatures during video recording? Temperature can affect skin color.

LINE 173: Please remove “…” after “size”.

LINES 175-189: This paragraph should be moved to the discussion section.

LINE 197: Please identify/describe the version of Python and script used to select the 30 random images.

LINE 199: Please explain how sampling was multiplied to minimize bias. How may times?

LINE 415: What was the high percentage?

LINES 418-419: Ruffled head feathers can also indicate sickness behavior. Be cautious with this interpretation.

LINES 428 and 429: Usually “mutual preening” is referred to as “allopreening”.

LINE 481: Ambient temperature needs to be reported in the materials and methods section, especially if this statement is going to be made.

LINE 515 and 516: Please combine these two sentences.

Reviewer #3: Overall comment:

The proposed manuscript is related to understanding emotionality in chickens using changes in facial expressions. This study, if replicated/repeated, could add valuable information in assessment of affective state as an indicator of bird welfare. However, the biggest limitation of this study is the fact that it is conducted in small number of birds in variable environmental conditions. The authors have added limitations of the study in the discussion section of the manuscript but without the knowledge of the ambient temperature and humidity conditions at two barns used in the study (at the very least), interpretation of the data cannot be complete.

Specific comments:

Material and methods:

- L119-120: Provide evidence to the prosocial claims of the hens used in the study.

- Provide temperature/humidity conditions between both farms.

- L137: Elaborate what ‘activation of white balance function’ does to the results of the study.

- L176: provide context behind ‘alert’ behavior observed in the methods section as well. Was ‘alert’ behavior displayed in response to the actual presence of the Common buzzard?

- L178: consider changing ‘affective state’ to ‘emotional state’

Discussion:

- L437-447: We have to be careful about the assumption that dustbathing behavior is rewarding to hens in absence of neurophysiological data from hens performing dustbathing. Could the discussion also explore the possibility that dustbathing yielded intermediate results because the behavior elicits neutral valence rather than the presumed positive valence?

- L481-482: This discussion should include at the very best ambient temperature and humidity data from both farms (M and P house). Also, there are studies to suggest changes in facial surface temperatures under different thermal conditions (For example - Kim, N. Y., Kim, S. J., Oh, M., Jang, S. Y., and Moon, S. H. 2021. Changes in facial surface temperature of laying hens under different thermal conditions. Anim. Biosci. 34(7): 1235-1242.). As this study was conducted in barn-type environment with less control over temperature and humidity, the effects of these factors on the results cannot be negated.

6. PLOS authors have the option to publish the peer review history of their article (what does this mean?). If published, this will include your full peer review and any attached files.

Reviewer #1: No

Reviewer #2: No

Reviewer #3: No

---

## [Author Response · Author response to Decision Letter 0]

21 May 2024

Dear Editor,

Please find below our responses to each point raised by the reviewers. We would like to thank the reviewers for their positive comments on our work. We think that their comments helped us to improve our manuscript.

The line numbers below refer to “Revised_Manuscript_with_Track_Changes”

Reviewer #1: The authors addressed the assessment of emotions in different contexts (valence positive or negative /arousal low or high) in domestic fowl by using new behavioural indicators in domestic birds: facial blushing and feather fluffing.

The study is creative and shows interesting results which might be a first step to find applications for improving to assess welfare.

The paper is well written, so apart from minor comments detailed below.

Comments in details :

No key words?

Key words were not asked for the submission. Key words proposed if required are: Facial expression, welfare, sentience, emotion, communication, Gallus gallus domesticus 

ABSTRACT:

Few information about results for feather fluffing. Is it possible to add little bit more?

We added (lines 29-31): “Feather position also varied with the situations. Feather fluffling was mostly observed in positively valenced situations, except when hens were eating.”

INTRODUCTION

L47: Replace our with the.

Done line 49

L61: Could you modify the sentence? The question is more that ethical question, is a welfare question.

Yes, it is true. We have replaced the sentence with: “This is critical when looking at bird welfare, particularly in case of domestic species. It is also an ethical question.” (lines 63-65)

MATERIAL and METHODS

You studied 2 strains : Pekin and Meusienne with respectively 10 and 8 hens. Is it a sample of hens within each flock? if it is the case need to be mentioned and add information how you selected the hens.

It was not a sample of hens. No more hens were available. They represent the total number of hens that we were able to obtain for each strain at the sites where we worked. For the Meusienne group, we had planned to work with a larger number of hens, but the eggs were collected in a remote location and, unfortunately, problems arose during egg transport due to the COVID pandemic situation at that time. 

To clarify that point, we added in the “Animals and sites” section: “These hens hatched on the farm and were reared in groups of mixed sexes and breeds before the experiments. Only ten female Pekin bantam from the same age and only 8 females Meusienne were available for the experiment. For both breeds, the experimental group was constituted one week before the beginning of the experiment.” (lines 125-128)

L121 “ ..known for their calmness and closeness with humans.” add bibliographic reference.

Two references have been added line 129 and lines 692-695.

- Husson H. Guide des races de poules. 130 races françaises et étrangères. Paris : Editions Ulmer;. 2019. pp. 232-233. 

- Nuttall P. Le guide Larousse des poules et du poulailler. Bien les choisir, les nourrir et les garder en bonne santé. Edition Larousse; 2020. p. 218. 

I am not sure to understand how you proceed to organize your sequence for filming.

Did you decide to follow one specific animal? Did you wait for the specific routine behaviours?

Need more information about your method.

We wait for specific routine behaviours. The objective was to limit as much as we could our movements to minimise bird disturbance. 

We have added some precisions about our method in the “Filming the hens” section (lines 165-168): “The observers waited for the specific routine behaviours to be expressed rather than filming a single animal for a given time. This method had the advantage of minimising the movements of the observers and therefore disturbances of the bird. It also allowed filming only when the hen’s head was clearly visible.”

L162: “hens spontaneously entered the arena”. No training or motivation stimulus to lead the animal to go in the arena? you wait all the day each animal entered in the arena? What happen with a hen which doesn’t to go inside. 

Add information about that.

Animals were very familiar with the observers. We added information how we proceed to obtain a good relationship with the hens (lines 148-150).

As mentioned in our previous manuscript (lines 184-186 in the revised manuscript), hens had been trained: “All hens had free access to the arena from approximately 36 hours before being tested. They had also been previously familiarized to eat mealworms from the dish inside the arena.”

In both groups, birds were waiting at the entrance of the arena. One of the difficulties was to be sure that only one hen entered. Those that had been already tested tried to enter again. For very few birds, one observer had to encourage them to the entrance by pushing their bottom gently when the other observer attracted the other hens. Tests were performed in 70 min then 50 min in P-hens and 50 min then 45 min in M-hens. 

Lines 187-189 we added: “All birds were waiting at the entrance of the arena. It took less than 70 minutes to test all P-hens and less than 50 minutes to test all M-hens.” 

L150: hens run fast, they didn’t try to escape when you tried to catch them ? How long did you wait before to catch the next one? each capture can have an impact on the all flock.

The first hens captured did not try to escape, but the following tried. We added lines 173: “When necessary, the hen was gently led to a corner of the field to be captured.” 

It is true that each capture has an impact on the flock. We did not wait before to catch the next one to test all the birds on a short period. The delay between each capture varied between 1 and 4 minutes for P-hens and 2 and 5 minutes for M-hens, depending on the delay necessary to capture the next hen. 

We added these precisions lines 177-179: “After one hen had been filmed, another one was captured. The delay to capture the next hen varied from 1 to 4 minutes in P-hens (mean = 2 min 28 s) and 2 to 5 minutes (mean = 3 min 30 s) in M-hens”. 

L174 No rings for P-hens?

It was not necessary. Their identification was easy with their phenotypic characters. We added why we used ring for M-hens lines 203: “because their plumage colorations were very similar.”

Image extraction, selection and analysis:

For some animals and situations, you got few images. What could be the impact on your results?

The impact is likely low. We worked on 165 to 300 (P-hens) or 112 to 212 (M-hens) images per situation, with a mean number per hen higher than 20 images per situation, except for alert where the mean number of images was 16.5 (P-hens) and 16.0 (M-hens). Using a random sampling on our images, we observed that the use of only 20 images per bird and situation to assess skin redness was sufficient to obtain similar results*. In the Alert situation were the number of images obtained was the lowest, the number of images per hen is higher than 20 in most of the cases (P-hens: 6/10 for redness and 6/8 for feather position, and M-hens: 4/7). 

* It is why in Soulet et al, 2024 (Applied Animal Behaviour Science, ttps://doi.org/10.1016/j.applanim.2024.106268) we used only 20 images per situation for each hen. Results obtained on 6 Sussex hens have confirmed our present results. 

We have added some precisions in the manuscript lines 240-244: “We used 1710 images for P-hens (165 to 300 images per situation) and 1336 images for M-hens (112 to 210 images per situation). For some behaviours and hens the number… (S1 Table), however similar results were obtained when analyses were performed on a random sampling of 15 to 30 images per hen and situation”. 

The figures below illustrate the percentage of cases (out of 1000 simulations carried out on our dataset – random sampling) where a significant difference was obtained (Wilcoxon test) between the skin redness values with n images (1<n<29) per hen and the 30 images per hen. Statistical analyses were applied on the median redness for each hen as in our study (see Statistical analysis section).

L198: “Images came, as far as possible, from different days and behavioural sequences”. What does it mean more information.

We clarified that point in lines 230-234. 

It seems that this part on the selection of the images was unclear since Reviewer #2 also had questions on this point (LINE 199). The paragraph has been modified lines 227-237 to answer the comments of both reviewers. We hope the information added clarify the method we used. Please let us know if it is not clear enough. 

At which time the image is selected for each behaviour ? at the beginning, in the middle , at the end ?

For example dustbathing last quite a long time and I guess the redness progessed over the time.

The selection was only based on three criteria that are mentioned lines 223-226 (193-196 in the previous manuscript): skin of comb, wattle, ear lobe and cheek visible on the image, uniform light and no direct sun on these skin areas. We added to clarify: “Selection was only based on these three criteria.” (lines 226-227). Even if some behaviours, such as dustbathing, last quite a long time, our objective was only to compare situations. It is difficult to have such a precision when selecting images (during dustbathing, birds are moving a lot, turn around, heads are often only partly visible, and as birds performed this behavior in group head is often hidden by a conspecific). 

As mentioned in the Discussion (lines 488-493 in the previous manuscript), dustbathing is characterized by different phases. Redness do not progress linearly but “We observed alternation of phases of high and low redness (as well different level of feather erection), likely due to an alternation of high and low arousal.” (lines 492-494 in the revised manuscript).

L271 : put ad libitum in italic

Done (line 311)

RESULTS

L291-292: The chi² analyze is not mentioned in statistical section. Please mention it.

As mentioned in statistical section lines 274-275, we used “a binomial Generalized Linear Mixed Model (GLMM) (lme4 package [40])”. In binomial GLMs, variables are tested for significance using the chi2 distribution (instead of the Fisher distribution in the Gaussian case), hence the Chi² statistic presented in the text. However, no Chi² test on a contingency table was performed.

L324 -337: Please write a sentence for the 3 others ROI in different situations.

Done lines 376-377. We added “For the three other ROI, results were similar but situations were more difficult to discriminate statistically, especially for wattles and comb.”

REFERENCES

L715: please correct this reference.

Done (lines 797-800)

Reviewer #2: This is a very interesting article investigating the redness of skin as a measure of valence and arousal in chickens. My edits and comments are detailed by line number below.

Abstract – Please add at least one statement about the results related to feather position and whether that was or was not repeatably indicative of affective state and why.

Done lines 29-31: “Feather position also varied with the situations. Feather fluffing was mostly observed in positively valanced situations, except when hens were eating.”

LINE 22: Please revise “domestic hens” to “female chickens” here and throughout. “Hen” can refer to females from many different domesticated poultry species.

LINE 84: See LINE 22.

Revised throughout the manuscript. Domestic hens have been replaced by “female chickens” (lines 116, 121, 427, 501, 508), “female domestic fowl” (lines 467, 473) or “domestic fowl” (lines 22, 238) depending on the sentences.

LINE 88: Please revise “rustic hens” to “hens from two rustic chicken breeds”.

Done line 92.

LINE 128: What were the outdoor temperatures during video recording? Temperature can affect skin color.

This is an interesting point. We added lines 193-194 “The outdoor temperatures were approximately 25°C (P-hens) and 27°C (M-hens) during the video recordings, and the weather was dry and mainly sunny.” and we have discussed this point more thoroughly than before in the Discussion section. See also answer to Reviewer #3, L481-482, for references and to complete the answer.

We found no references showing that temperature can affect skin colour (or can induce rapid changes in skin colour). The only reference we found concerned the effect of the temperature on skin temperature measured by infrared thermography. However, it seems there is no direct link between redness and skin temperature. In the situations where skin redness was high in our study (capture, reward), skin temperature decreases in the literature (when hens are tested in similar situations). See answer to Reviewer #3, L481-482. 

We did not register ambient temperature precisely (hour by hour throughout a day) as we noticed no effect of the temperature on skin redness of our birds. As mentioned lines 533-534 and shown on the video (supplementary material S3), the phenomenon of facial skin flushing is transient. The colour of the face can redden or lighten in a few seconds or minutes depending on the duration of the stimulation. As ambient temperature does not vary within these short laps of time it cannot explain the differences observed. To illustrate this point, you can find below pictures of the same bird at rest (left) and during the capture test (right) taken approximately 15 minutes apart. The sentence is now: “… the blushing measured in our study appears in few seconds and is transitory. It lasts a few seconds or minutes depending on the duration of the stimulus inducing the reaction (a video can be seen on S3 Video), contrary to temperature variations during a day.” (lines 533-536).

To consider the potential effects induced by working on birds living outside in semi-natural conditions (including weather variations), we took several precautions. It seems that they did not appear clearly enough in our manuscript.

1- Temperature variations were limited when films were performed. Information added line 192: “Films were performed between 10:00 and 18:00 to avoid sunrise and sunset (at this season the sun is high in the sky and temperature variations are limited;...” 

2- All the situations were filmed both in the morning and the afternoon and across the successive days of the observational period (3 weeks). “Each routine behaviour (situations) was filmed both in the morning and the afternoon and across the successive days of the observational period” added lines 168-169. 

- We worked on 30 images for each hen and situation which means 100 to 300 images per situations. Precisions on the methods added lines 227-237 and 240-244.

LINE 173: Please remove “…” after “size”.

Done line 202.

LINES 175-189: This paragraph should be moved to the discussion section.

We think it is necessary to explain why these situations were selected for this study (see for example the comment Reviewer #3 L437-447). As this change was not required by the two other reviewers, we preferred not to move this paragraph to the discussion section. 

LINE 197: Please identify/describe the version of Python and script used to select the 30 random images.

The script is freely available on https://forgemia.inra.fr/projetred/red_project (sampling_files.py”). This information had been added line 229. We used Python version 3.8 

LINE 199: Please explain how sampling was multiplied to minimize bias. How may times?

It referred to the 30 images for each hen and situation. As this part on images selection appeared unclear (see also Reviewer #1 L198 comment), it has been rewritten to clarify (lines 226-237). 

LINE 415: What was the high percentage?

Values of the medians (provided Table 1) have been added lines 455-456: “… (median at 40 % or more in M-hens and 56 % or more in P-hens).”

LINES 418-419: Ruffled head feathers can also indicate sickness behavior. Be cautious with this interpretation.

Yes, it is true. We have modified our sentence and included this information lines 459-462: “Fluffing of head feathers appears to be a relevant indicator of positive emotional states in situation where birds are healthy. Indeed, feather fluffing could also signal sickness in birds (in this case, birds are often prostrated and fe

---

## [Decision Letter · Decision Letter 1]

21 Jun 2024

Facial blushing and feather fluffing are indicators of emotions in domestic fowl (Gallus gallus domesticus )

PONE-D-23-42910R1

Dear Dr. Arnould,

We’re pleased to inform you that your manuscript has been judged scientifically suitable for publication and will be formally accepted for publication once it meets all outstanding technical requirements.

Kind regards,

Birendra Mishra, DVM, PhD

Academic Editor

PLOS ONE

Additional Editor Comments (optional):

The authors have addressed all the comments from the reviewers and the editor.

Reviewers' comments:

Reviewer's Responses to Questions

**Comments to the Author**

1. If the authors have adequately addressed your comments raised in a previous round of review and you feel that this manuscript is now acceptable for publication, you may indicate that here to bypass the “Comments to the Author” section, enter your conflict of interest statement in the “Confidential to Editor” section, and submit your "Accept" recommendation.

Reviewer #2: All comments have been addressed

Reviewer #3: (No Response)

2. Is the manuscript technically sound, and do the data support the conclusions?

Reviewer #2: Yes

Reviewer #3: (No Response)

3. Has the statistical analysis been performed appropriately and rigorously? 

Reviewer #2: Yes

Reviewer #3: (No Response)

4. Have the authors made all data underlying the findings in their manuscript fully available?

Reviewer #2: Yes

Reviewer #3: (No Response)

5. Is the manuscript presented in an intelligible fashion and written in standard English?

Reviewer #2: Yes

Reviewer #3: (No Response)

6. Review Comments to the Author

Reviewer #2: I am satisfied with how the authors addressed my review and believe the journal should accept the revised manuscript in its current form.

Reviewer #3: (No Response)

7. PLOS authors have the option to publish the peer review history of their article (what does this mean?). If published, this will include your full peer review and any attached files.

Reviewer #2: No

Reviewer #3: **Yes: **Prafulla Regmi

---

## [Editor Report · Acceptance letter]

2 Jul 2024

PONE-D-23-42910R1 

PLOS ONE

Dear Dr. Arnould, 

I'm pleased to inform you that your manuscript has been deemed suitable for publication in PLOS ONE. Congratulations! Your manuscript is now being handed over to our production team.

Kind regards, 

on behalf of

Dr. Birendra Mishra 

Academic Editor

PLOS ONE